# Generalizing Neural Additive Models via Statistical Multimodal Analysis

**Young Kyung Kim** [1]  **J. Matías Di Martino** [1]  **Guillermo Sapiro** [1]

## Abstract

Generalized Additive Models (GAM) (Hastie, 2017) and Neural Additive Models (NAM) (Agarwal et al., 2021) have gained a lot of attention for addressing trade-offs between accuracy and interpretability of machine learning models. Although the field has focused on minimizing trade-offs between accuracy and interpretability, the limitation of GAM or NAM on data that has multiple subpopulations, differentiated by latent variables with distinctive relationships between features and outputs, has rarely been addressed. The main reason behind this limitation is that these models collapse multiple relationships by being forced to fit the data in a unimodal fashion. Here, we address and describe the overlooked limitation of "one-fits-all" interpretable methods and propose a Mixture of Neural Additive Models (MNAM) to overcome it. The proposed MNAM learns relationships between features and outputs in a multimodal fashion and assigns a probability to each mode. Based on a subpopulation, MNAM will activate one or more matching modes by increasing their probability. Thus, the objective of MNAM is to learn multiple relationships and activate the right relationships by automatically identifying subpopulations of interest. Similar to how GAM and NAM have fixed relationships between features and outputs, MNAM will maintain interpretability by having multiple fixed relationships. We demonstrate how the proposed MNAM balances between rich representations and interpretability with numerous empirical observations and pedagogical studies. The code is available at (https://github.com/youngkyungkim93/MNAM).

---
[1]Duke University. Correspondence to: Young Kyung Kim <yk206@duke.edu>, J. Matías Di Martino <matias.di.martino@duke.edu>, Guillermo Sapiro <guillermo.sapiro@duke.edu>.

*Workshop on Interpretable ML in Healthcare at International Conference on Machine Learning (ICML)*, Honolulu, Hawaii, USA. 2023. Copyright 2023 by the author(s).

## 1. Introduction

Deep neural networks (DNN) achieve extraordinary results across several important applications such as object detection (Redmon et al., 2016; Girshick et al., 2014; Ren et al., 2015), object classification (He et al., 2016; Krizhevsky et al., 2017; Dosovitskiy et al., 2020), and natural language processing (Mikolov et al., 2013; Devlin et al., 2018; Brown et al., 2020). Yet DNN's popularity is still low in critical applications where miss-classification has high consequences or transparency is required for decision-making, e.g., to prevent unfairness toward certain groups; examples are medical-related risk estimation and machine learning (ML) based public policies. According to experts in these domains, one of the main factors limiting the adoption of DNN-based approaches is the lack of interpretability and trustworthiness associated with these algorithms (Shorten et al., 2021; Amarasinghe et al., 2020; Li et al., 2022). Even though several techniques have been proposed to increase the understanding of DNN (Agarwal et al., 2021; Ribeiro et al., 2016; Pedapati et al., 2020), medical professionals or policymakers still prefer simple models for which they can understand directly the factors that lead to a particular prediction. On the opposite end of DNN are algorithms such as linear regression and its multiple variants (Montgomery et al., 2021), which are simple and interpretable but lack the flexibility and high performance that DNN has. Notably, linear models can't capture nonlinear relationships and can't exploit numerous novel tools that efficiently optimize modern DNN approaches. A recent approach proposed by Agarwal et al., named Neural Additive Models (NAM), which is a form of Generalized Additive Models (GAM), provides an interesting balance between interpretability and learning power. Individual features undergo nonlinear transformations independently, and these transformed features are merged in a regression-like paradigm, allowing the user to understand the weight of each factor leading to a prediction. This enables the algorithm to learn non-trivial relationships between the features and the target outcomes while leveraging powerful state-of-the-art optimization tools developed for deep learning.

Although most of the research on GAM has focused on minimizing the trade-offs between accuracy and interpretability (Nori et al., 2019; Zuur, 2012; Agarwal et al., 2021),

addressing the lack of power for GAM and NAM in capturing multimodal relationships between input and target variables has been rare or nonexistent. This limitation is crucial especially when a dataset has multiple relationships with distinctive relationships between features and outputs. For example, imaging in the context of a medical application where we are predicting the glucose level $y$ using electronic health records (EHR) as input variables $x_1, ..., x_n$; let us assume there are two subpopulations identified by the variable $d \in \{0, 1\}$, which can be observed or latent features. For both cases, NAM would fail to capture a relationship in which $y$ is positively correlated with $(x_1|d = 0)$ but is uncorrelated with $(x_1|d = 1)$. This is due to NAM only learning one deterministic relationship between input and output. When $d$ is a latent variable, NAM will fail to differentiate them and collapse two relationships into one by averaging them to learn one deterministic relationship. Even if $d$ is an observed feature, NAM will fail to differentiate them as a DNN assigned for $X_1$ doesn't take $d$ as an input to have information on two subpopulations.

To address this while preserving the virtues of NAM, we propose a probabilistic Mixture of Neural Additive Models (MNAM). The main idea is to apply mixture density networks (MDNs), a neural network with mixture of $k$ Gaussian distributions as an outcome, as a linking function for GAM to model the relationship between input and outcome in a multimodal relationship and associate a probability to each mode. The probability of each mode enables the model to be flexible in representing multiple subpopulations as MNAM is able to activate accurate relationships for certain subpopulations by increasing their probability.

Figure 1 illustrates the power and flexibility of MNAM. These strengths are also illustrated in Section 3 through applying MNAM on real datasets. Such flexibility will be especially crucial in decision-making with high consequences. For example, for analyzing the side effects of medicine, 99% of participants might have steady glucose levels but 1% might have high and dangerous glucose levels after taking a medicine. NAM will collapse both levels into one indicating no side effects on average, but MNAM will accurately show, with probability, two glucose levels of different subpopulations.

It is important to highlight the interpretability of the model. Similar to NAM having a one fixed relationship between input features and output variables, MNAM will have fixed multiple relationships, which makes the model interpretable. Only the probability of each mode will change from the change in other features, which indicates changes in a subpopulation. Finally, just as for NAM, all powerful state-of-the-art tools developed for deep learning are applicable to MNAM.

Our main contributions are: (i) we identify the overlooked

limitation that GAM and NAM have when they are trained with a dataset that has multiple subpopulations; (ii) we provides a practical alternative to solve the critical problem or "one-fits-all" standard in interpretable DL approaches; (iii) we propose a model called MNAM that could learn multiple relationships among subpopulations for the solution; (iv) we propose a method to train MNAM, with objectives to learn multiple relationships and activate one or more matching relationships for a given subpopulation; and (v) we demonstrate MNAM is more expressive in accuracy and flexible in interpretability compared to NAM. We describe the proposed method in Section 2. Section 3 presents empirical evidence and pedagogical studies, showing strengths of MNAM. We discuss related work in Section 4 and limitations in Section 5. Finally, we provide a conclusion in Section 6.

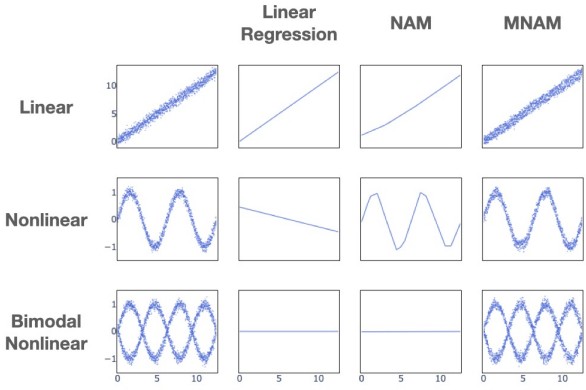

*Figure 1.* Linear regression, NAM, and MNAM on linear, nonlinear, and bimodal data. The left column illustrates the input for three datasets. The columns illustrate the representations learned by linear regression, NAM, and MNAM, respectively. As expected, linear regression fails to learn datasets with nonlinear relationships. NAM fails to learn datasets with relationships that have more than one modality, and only MNAM is able to learn nonlinear and multimodal relationships.

## 2. Method

### 2.1. Problem Statement

The GAM and NAM models lack the ability to effectively represent multimodal relationships between input and output due to their link functions. These link functions have a single output, which results in the averaging of the multimodal relationship to minimize loss. To address this limitation, we require an additive model that utilizes a function with multiple outputs for link functions. By training each output to represent a specific mode, rather than learning the average of the multimodal relationship, we can overcome

this limitation. Furthermore, GAN and NAM models also fall short in identifying subpopulations, as their main objective is to learn the average of multimodal relationships rather than learning each mode separately. To overcome this limitation, an additional model is needed to learn the identification of different subpopulations and activate the appropriate mode or outcome within the additive model with multiple outcomes.

To address both of these challenges, the MNAM model has been developed. MNAM tackles these problems by incorporating a function with multiple outputs for link functions, enabling the representation of multimodal relationships, and by incorporating a separate model to identify subpopulations and activate the relevant mode within the additive model.

## 2.2. Architecture

In order to represent the multimodal relationship between inputs and outcomes, MNAM has an outcome of a mixture of $k$ Gaussian distributions, which are described as $(\mathcal{N}_1(\mu_1, \sigma_1^2), ..., \mathcal{N}_k(\mu_k, \sigma_k^2), \pi_1, ..., \pi_k)$. $\mathcal{N}_i(\mu_i, \sigma_i^2)$ denotes the standard Gaussian distribution with $\mu_i$ as mean and $\sigma_i$ as standard deviation, while $\pi_i$ represents the probability associated with it. Since the Gaussian mixture model is a universal approximator for any density distributions, MNAM will be able to approximate any multimodal relationships given large enough $k$. We formalize this notion in Section 3.2. One or more Gaussian distributions will be assigned to one of the input-output relationships for the representation and MNAM will activate certain relationships for given subpopulations of the input by increasing the probability of the appropriate Gaussian distributions. This is an important property as it indicates that we can successfully capture and represent modes for relationships on various subpopulations in the dataset, without knowing the number of modes in advance. Such property will be shown in Section 3.2 through a pedagogical example.

Similar to NAM, MNAM predictions are built from a linear combination of embeddings $Z_i$ of each input feature $X_i$ mapped through a neural network. In contrast with NAM, MNAM embedding consists of parameters for $k$ Gaussian distributions and a latent variable for predicting the probability of the mixture of $k$ Gaussian distribution models $(\mathcal{N}_{1,j}(\mu_{1,j}, \sigma_{1,j}^2), ..., \mathcal{N}_{k,j}(\mu_{k,j}, \sigma_{k,j}^2), Z_j^\pi)$. The left index of the Gaussian distributions is a reference to the number of components for the mixture and the right index $j$ is a reference to one of the input features. As shown in Equation 1, we compute the mean and variance of the Gaussian distributions for the MNAM outcome by linearly combining the mean and variance of matching components for Gaussian distributions of features' embedding.

$$\mathcal{N}_{i,1}(\mu_{i,1}, \sigma_{i,1}^2) + ... + \mathcal{N}_{i,m}(\mu_{i,m}, \sigma_{i,m}^2)$$
$$= \mathcal{N}(\sum_{j=1}^{m} \mu_{i,j}, \sum_{j=1}^{m} \sigma_{i,j}^2) = \mathcal{N}_i(\mu_i, \sigma_i^2) \quad (1)$$

The advantage of this linear property of summation for the Gaussian distributions is that MNAM is able to linearly represent how much the overall mean and uncertainty of prediction changes due to changes in a feature.

Latent variables for predicting the probability of the mixture of $k$ Gaussian distributions for all features' embeddings will be the input for a separate neural network that predicts the probability of the output. This neural network will learn to identify which subpopulation is being represented based on input from all features, and activate the correct relationships by assigning a high probability to the matching Gaussian distributions. The description of how MNAM computes predictions is summarized in Appendix A and the comparison of the architecture for NAM and MNAM is illustrated in Figure 2.

## 2.3. Training and Optimization

As mentioned in Section 1, state-of-the-art optimization tools for deep learning are applicable for training MNAM. For this work, we used Adam (Kingma & Ba, 2014) with a learning rate decreasing by 0.5% for each epoch. The objective of the training and optimization of MNAM is to assign one or more Gaussian distributions to each relationship in the dataset. Another objective is to learn to identify subpopulations from the given features to activate the correct relationship associated with the given sample. We devise hard-thresholding (HT) and soft-thresholding (ST) algorithms for the given objectives. The HT algorithm trains or updates a single mode or a Gaussian distribution with a minimum loss, while the ST algorithm trains or updates all $k$ modes or Gaussian distribution with weights computed by the likelihood of each mode on the label. Between the two algorithms, we chose the HT algorithm because the algorithm is more numerically stable and computationally efficient compared to the ST model, which has been shown in the empirical experiment in Section B in the appendix. The detailed description of the HT algorithm is described next. The detailed description of the ST algorithm can be found in Section B in the appendix.

### 2.3.1. HARD-THRESHOLDING (HT) ALGORITHM

Given the output of a mixture of $k$ Gaussian distributions for MNAM, the Gaussian negative log-likelihood (GNLL) loss is computed for each Gaussian distribution against a label. Among $k$ losses, only the minimum factor will be used to compute the total loss, which means only weights used to

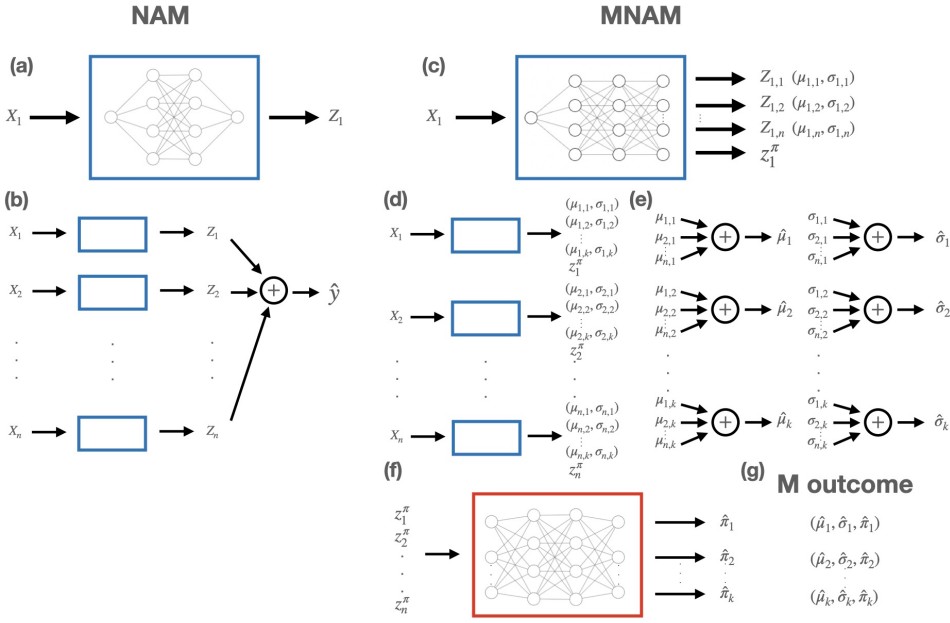

*Figure 2.* Illustrative schemes of NAM and MNAM network architectures. As shown in (a) and (b), NAM independently maps features into embedding through neural networks and then linearly combines embeddings for a prediction. Similar to NAM, MNAM independently maps features into embeddings through neural networks. The difference is that embedding consists of $k$ Gaussian distributions and a latent variable for predicting probabilities for a mixture of the $k$ Gaussian distributions, which is illustrated in (c) and (d). (e) illustrates linear combinations of each component of the Gaussian distributions for all features' embeddings. (f) depicts the mapping of latent variables for a mixture of $k$ Gaussian distributions ($Z_1^\pi, Z_2^\pi, ..., Z_n^\pi$) into probabilities for the mixture of $k$ Gaussian distributions through a neural network. (g) is an example of the outcome for MNAM, which is the mixture of $k$ Gaussian distributions.

compute minimum loss are updated from a backpropagation. This enables the model to assign one Gaussian distribution to learn each relationship. Cross-entropy loss between the probabilities of a mixture of Gaussian distributions for a prediction and the index number of the Gaussian distribution with the minimum loss is computed to measure how well MNAM activates the corresponding Gaussian distribution for the input. This loss enables the model to learn to identify subpopulations for a given input to increase the probability of the correct Gaussian distribution for representation. Algorithm 1 summarizes the proposed training algorithm. It is important to highlight that the proposed learning method is unsupervised, in the sense that the data subgroups do not need to be known or defined in advance.

---

**Algorithm 1** Hard-Thresholding (HT) Algorithm

**Input: Data $(X, Y)$, MNAM $f$, GNLL loss $g$, Cross-entropy loss function $h$, Rate for cross-entropy loss $\lambda$**

$\mathcal{N}_1(\mu_1, \sigma_1), ..., \mathcal{N}_k(\mu_k, \sigma_k), \pi_1, ..., \pi_k = f(X)$
$min\_loss = 0$
**for** $i = 1$ **to** $k$ **do**
    $gau\_loss = g(\mathcal{N}_i(\mu_i, \sigma_i), Y)$
    **if** $min\_loss > gau\_loss$ **then**
        $min\_loss = gau\_loss$
        $min\_index = i$
    **end if**
**end for**
$prob\_loss = h((\pi_1, ..., \pi_k), min\_index)$
$total\_loss = Min\_loss + \lambda \cdot prob\_loss$

---

## 3. Result

### 3.1. Empirical Observations

#### 3.1.1. DATASETS

We evaluate six datasets: the California Housing (CA Housing) (Pace & Barry, 1997), the Fair Isaac Corporation (FICO) (FICO, 2018), the New York Citi Bike (BIKE) (Vanschoren et al., 2013), the Medical Information Mart for Intensive Care (MIMIC-III) (Johnson et al., 2016), the US Census data on Income (ACS Income) for California in 2018 (Ding et al., 2021), and the US Census data on Travel time (ACS Travel) for California in 2018 (Ding et al., 2021).

### 3.1.2. Training and Evaluation

Similar to how the original paper trained NAM, we used Bayesian optimization (Močkus, 1975) to finetune variables to train NAM and MNAM. Learning rate, weight decay, and output penalty are finetuned for NAM. Learning rate, weight decay, output penalty, number of Gaussian distributions, and lambda for cross-entropy loss are finetuned for MNAM. For both models, we utilized early stopping to reduce overfitting. Optimized parameters from Bayesian optimization can be found in the table from Section C in the appendix. We used a 5-fold cross-validation for CA Housing, FICO, and MIMIC-III datasets, and a 3-fold cross-validation for BIKE, ACS Income, and ACS Travel datasets. For evaluation, we trained 20 different models by randomly splitting the train set into train and validation sets for each fold. We ensembled 20 models to evaluate on the test set.

In comparing deterministic and probabilistic models, we encountered a challenge due to the lack of standardized evaluation metrics. Therefore, we decided to use the mean absolute error (MAE) as a metric for comparison. However, MAE has a limitation. It fails to account for the uncertainty in predictions made by probabilistic models. Even if a probabilistic model accurately predicts a true distribution for the label distribution, it may still receive the same MAE score as a deterministic model if it is correct in predicting the mean of the label distribution. To address this limitation, we transformed NAM into a probabilistic model (pNAM) by setting k=1 in MNAM. We then utilized likelihood as a metric to compare the performance of pNAM with the remaining deterministic models, to emphasize the importance of having multimodal compared to unimodal distribution as an outcome. For the remaining deterministic models, we used the earth mover's distance (EMD) to assess how well they learned to approximate the label distribution. The EMD scores of models can be found in the table from Section E in the appendix. It's worth noting that the EMD score is an unfair evaluation for deterministic models. This is because deterministic models do not learn the uncertainty of the data during training, unlike probabilistic models.

### 3.1.3. Regularization

Similar to NAM, all regularization methods for deep learning can be applied to MNAM, including weight decay, dropout, and output penalty. For this study, we utilized weight decay and output penalty.

### 3.1.4. Results

Table 1 displays the MAE scores of NAM and MNAM, as well as the likelihood scores of pNAM and MNAM on datasets described above. MNAM consistently exhibited similar or superior MAE scores compared to NAM across all six datasets. Moreover, MNAM showcased a signifi-

cantly improved performance in terms of likelihood scores when compared to NAM for all datasets, except for the FICO dataset. Notably, the optimized number of Gaussian distributions for MNAM was 1, which means that pNAM and MNAM are identical models. This finding underscores MNAM's remarkable ability to effectively learn the output distribution, surpassing both NAM and pNAM in this aspect.

Differences in performance between MNAM and NAM differ greatly by datasets. Specifically, the discrepancy in likelihood scores between NAM and MNAM is much more pronounced for the CA Housing dataset compared to the ACS Income dataset. Several explanations could account for these observations. Firstly, the CA Housing dataset might exhibit more intricate interaction relationships among its features, rendering it more challenging for NAM to accurately capture the underlying patterns without any interaction term learning. Conversely, MNAM, with its enhanced capability to model complex interactions, would demonstrate an improved likelihood score on such datasets. Secondly, the CA Housing dataset might possess modes that differ more significantly from one another, making it harder to fit using a single Gaussian distribution for NAM. In this scenario, MNAM would enhance the likelihood score by accommodating the complexity of interaction relationships and the differences among modes within the datasets.

| | NAM | pNAM | MNAM | |
|---|---|---|---|---|
| Dataset | MAE↓ | Likelihood↑ | MAE↓ | Likelihood↑ |
| CA Housing | $0.48 \pm 9e^{-05}$ | $0.58 \pm 6e^{-04}$ | $0.46 \pm 4e^{-05}$ | $0.73 \pm 0.001$ |
| FICO | $2.7 \pm 0.002$ | $0.084 \pm 2e^{-06}$ | $2.7 \pm 0.002$ | $0.084 \pm 2e^{-06}$ |
| MIMIC | $1.5 \pm 0.0002$ | $0.15 \pm 3e^{-06}$ | $1.5 \pm 0.0003$ | $0.25 \pm 6e^{-05}$ |
| BIKE | $3.4 \pm 0.0005$ | $0.069 \pm 3e^{-08}$ | $3.4 \pm 0.0006$ | $0.092 \pm 1e^{-06}$ |
| ACS Income | $37.2 \pm 0.003$ | $0.011 \pm 4e^{-07}$ | $35.7 \pm 0.02$ | $0.013 \pm 4e^{-07}$ |
| ACS Travel | $15.6 \pm 0.0004$ | $0.017 \pm 2e^{-08}$ | $15.5 \pm 0.002$ | $0.036 \pm 2e^{-05}$ |

*Table 1.* MAE score for NAM and MNAM, and likelihood score for pNAM and MNAM on CA Housing, FICO, MIMIC, BIKE, ACS Income, and ACS Travel dataset

### 3.1.5. Out-of-Distribution Robustness

To evaluate the out-of-distribution robustness of pNAM and MNAM models, we trained the models on the 2018 California dataset of ACS Income and ACS Travel. We then computed the mean and variance of the likelihood scores for these models when evaluated on data from different states and years, ranging from 2014 to 2017, for both ACS Income and ACS Travel. The computation of the mean and variance is presented in Table 2. MNAM consistently outperformed pNAM in all four scenarios. Interestingly, both pNAM and MNAM demonstrated better or similar performance when evaluated on out-of-distribution datasets compared to in-distribution datasets. This could be attributed to the fact that pNAM and MNAM had a larger training set when evaluating on out-of-distribution data, as they didn't need to

reserve a third of the dataset for testing purposes. However, when it came to the state shift of ACS Income, the performance of both pNAM and MNAM significantly decreased compared to their performance on in-distribution evaluation. This indicates that there are greater variations among states for ACS Income compared to other out-of-distribution datasets. Nevertheless, the performance gap between pNAM and MNAM remained consistent across different datasets. This evaluation highlights the strength of MNAM in maintaining flexibility when dealing with out-of-distribution data compared to pNAM, showcasing its ability to adapt and capture nuanced relationships between variables.

| | ACS INCOME (LIKELIHOOD↑) | | ACS TRAVEL (LIKELIHOOD↑) | |
| --- | --- | --- | --- | --- |
| | STATE SHIFT | YEAR SHIFT | STATE SHIFT | YEAR SHIFT |
| pNAM | $0.0065 \pm 2e^{-07}$ | $0.011 \pm 2e^{-07}$ | $0.016 \pm 1e^{-06}$ | $0.017 \pm 3e^{-08}$ |
| MNAM | $0.0083 \pm 5e^{-07}$ | $0.015 \pm 5e^{-07}$ | $0.037 \pm 4e^{-05}$ | $0.035 \pm 3e^{-06}$ |

*Table 2.* Comparison of likelihood score for pNAM and MNAM, which are trained on 2018 California dataset of ACS Income and ACS Travel, evaluated on different states and years of ACS Income and ACS Travel.

### 3.1.6. INTERPRETABILITY

In this section, we visualize the relationships between features and labels, and how different relationships are activated from changes in subpopulations; we illustrate this for the MIMIC and CA Housing datasets. These examples illustrate the strength of the interpretability of MNAM. Relationships plots for all five datasets can be found in Section F in the appendix. As illustrated in the left column of Figure 3 and Figure 4, MNAM is able to learn and represent multiple relationships between features and labels, which NAM fails to do as it collapses those relationships into mean. Therefore, MNAM is more flexible in explaining and representing multiple relationships between features and labels by activating one or multiple of them.

Allowing multimodal data representations sheds light on non-trivial data relationships that are otherwise hidden in average "one-fit-all" models. For example, as illustrated in Figure 3, the variance or the discrimination of the length of stay among different ethnicities significantly differ between the two modes recognized by MNAM. (The left graph of Figure 3) The first relationship, which is a red line, has more variance or discrimination among ethnicities compared to the second relationship, which is a blue line, in the length of stay. If we group the algorithm's output by admission type (the middle graph represents common admission and the right graph represents urgent admission), we recognize the model activates more on relationships with less discrimination among ethnicities with urgent admission and vice versa with common admission. In other words, there is more variance and discrimination in the length of stay among ethnicities for common admission compared to urgent ad-

mission.

The strengths of MNAM is more evident when trained on the CA Housing dataset as the differences in activation of relationships among subpopulation are more drastic than when MNAM is trained on the MIMIC dataset. In Figure 4, we identified that the price of a house could increase or decrease as the number of people in the neighborhood increases (the first column of Figure 4, illustrates the two modes recognized by MNAM). If we group the algorithm's output by median income (the first row of the second column represents the bottom one percent, and the first row of the third column represents the top one percent), we can recognize that one of the modalities is associated with higher income households and the other with lower income households. For example, the first row of the second column shows that the top mode is activated more frequently on this subgroup (darker blue represents higher frequency), suggesting that the larger the number the people in the neighborhood, the higher the house prices. The opposite can be recognized for the higher-income subgroups (see the first row of the third column). In other words, the output of the model suggests that for wealthier neighborhoods, the more people, the less expensive houses are, while the opposite occurs in poor communities. A similar story is illustrated when we group the algorithm's output by proximity to the beach (the second row of the second column represents inland, and the second row of the third column represents the area near the beach). The output of the model suggests that for areas near beaches, the more people the more expensive houses are, while the opposite occurs inland. Notice how these rich data interpretations would have been missed using NAM, where a "one-fit-all" model is optimized.

### 3.1.7. COMPLEXITY AND TRAINING EFFICIENCY

Table 7 shows the comparison of average training time, training time per epoch, and the number of epochs required to train NAM and MNAM on different datasets. As expected, MNAM takes longer to train per epoch than NAM because it has an additional neural network for computing mode weights. Therefore, MNAM will be inefficient during prediction and evaluation compared to NAM as the model has higher computational complexity. However, interestingly, MNAM was faster than NAM in training for half of the datasets, and MNAM required fewer epochs than NAM for all datasets except one. Our hypothesis is that MNAM's assignment of modes to subpopulations effectively shrinks the space for the model to explore, resulting in fewer epochs needed for training, as each mode only needs to represent one subpopulation. In contrast, NAM has only one outcome that must represent multiple subpopulations, causing it to oscillate among subpopulations for representation during training. Overall, MNAM may be more efficient than NAM in terms of training time, despite having more parameters to

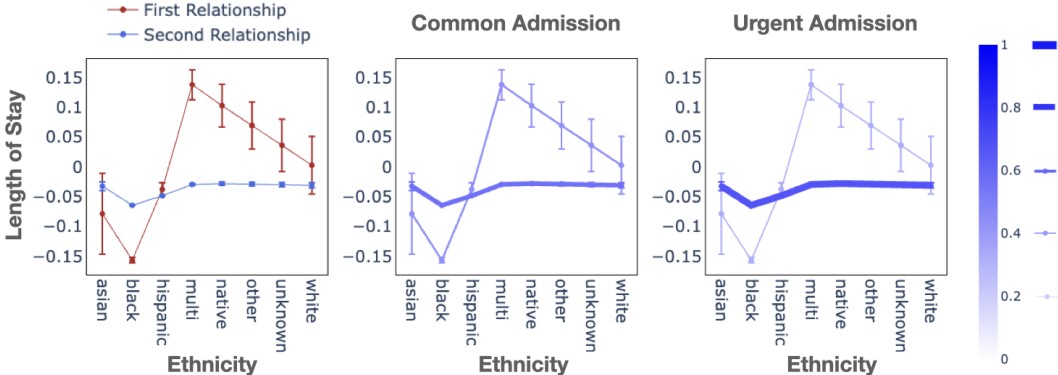

*Figure 3.* Line graphs on the relationship between the length of stay and the ethnicity for MIMIC with whisker representing a variance. The left graph represents two modes recognized by MNAM between the length of stay and the ethnicity. The middle and right graphs represent changes in the activation of two modes from changes in admission type. Except for admission type, all other remaining features have been fixed to mean values. The magnitude of the mode's activation is illustrated through the intensity of color and thickness of lines. The darker blue and thicker line represents higher activation of a mode. The blue color bar and different thickness of lines on the right side of the color bar represents the magnitude of a mode's activation.

compute.

| | TRAINING TIME (SECONDS)↓ | | TRAINING TIME PER EPOCH (SECONDS)↓ | | NUMBER OF EPOCHS↓ | |
|---|---|---|---|---|---|---|
| DATASET | NAM | MNAM | NAM | MNAM | NAM | MNAM |
| CA HOUSING | 636.38 | 411.73 | 0.72 | 0.84 | 885.36 | 489.85 |
| FICO | 326.26 | 337.54 | 0.78 | 0.85 | 417.34 | 397.67 |
| MIMIC | 173.10 | 278.58 | 0.93 | 1.09 | 186.64 | 256.53 |
| BIKE | 410.42 | 477.62 | 1.37 | 1.85 | 298.76 | 258.36 |
| ACS INCOME | 1460.31 | 896.41 | 3.22 | 4.47 | 453.65 | 200.40 |
| ACS TRAVEL | 2411.51 | 946.55 | 3.86 | 5.75 | 624.43 | 164.58 |

*Table 3.* Comparisons of average training time in seconds, training time per epoch in seconds, and number of epochs between MNAM and NAM

### 3.2. Pedagogical Example

For pedagogical value and to further illustrate the differences between the original NAM and the proposed MNAM, we created a synthetic dataset with different subpopulations, which are differentiated by either observed or latent variables. NAM has limitations in accurately representing such dataset as it collapses four relationships between $X_1$ and $Y$ into one deterministic relationship by averaging them. When $X_2$ is an observed variable, NAM is not able to differentiate relationships, since a neural network assigned to $X_1$ does not take $X_2$ as input. The neural network for $X_1$ simply uses the average relationship for representation, which is shown when $X_2 = 0$ and $X_2 = 1$. The representation is worsened for NAM when variables that differentiate subpopulations are latent variables, which is the case for $X_2 = 2$ and $X_2 = 3$ in the synthetic dataset. NAM tries to represent multiple relationships with one relationship, as shown in the second column of Figure 5. This can be critical, for example, in medical applications, where a drug might be effective in a certain subgroup of the population, tools like MNAM, would allow identifying from data modes or

outliers that might not fit the general expected therapeutic trend. MNAM overcomes such limitations as it is able to learn four relationships and activate the right relationships for each subpopulation. Another strength of MNAM is that as long as $k$ is larger than the number of relationships in a dataset, MNAM will be able to represent the relationships accurately. In other words, tuning $k$ is not critical, as long as its value is higher than the expected number of modes. Furthermore, MNAM is able to learn the uncertainty of each relationship, which NAM is unable to do. Described limitations of NAM and strengths of MNAM are illustrated in Figure 5.

### 3.3. Trade-offs between Accuracy and Interpretability

In this section, we compared different models to explore trade-offs between accuracy and interpretability. We evaluated Linear Regression (LR); NAM; the here proposed MNAM; Explainable Boosting Machine (EBM) (Nori et al., 2019), which is a form of Generalized Additive Models (GAM) with pairwise interaction terms; and Gradient Boosting Trees (GBT) (Friedman, 2001; Pedregosa et al., 2011). We used grid search for LR, EBM, and GBT to finetune hyperparameters for training. Table 4 shows the MAE scores for these five models. The order of the columns, left to right, represents an increase in complexity and a decrease in interpretability (here considered as a clear relationship between input and output). The table is split into two, which are models with direct relationships and complex relationships (left and right respectively). LR, NAM, and MNAM are models with direct relationships because their feature and output relationships are fixed even from changes in other features. Meanwhile, EBM and GBT are considered as models with complex relationships as their feature and output

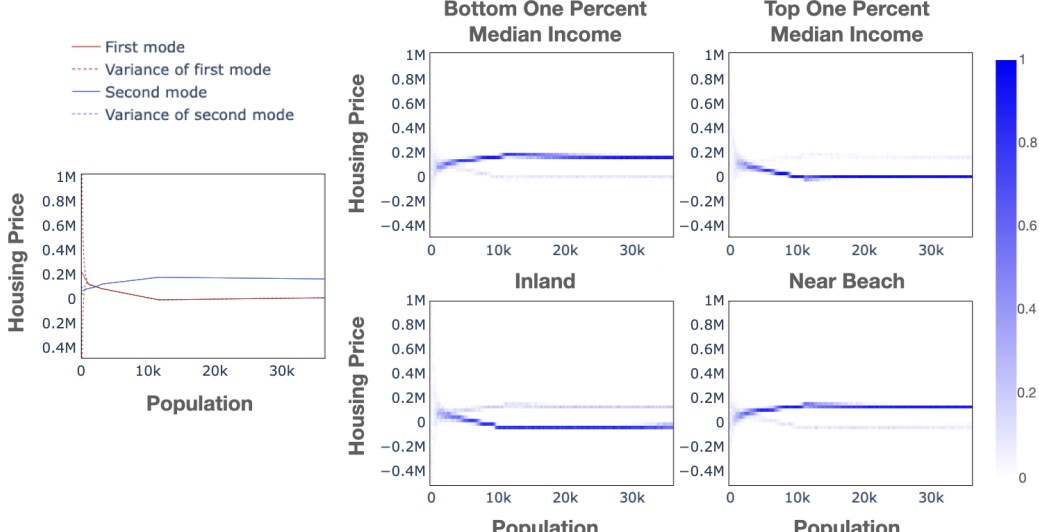

*Figure 4.* Line graph and heatmaps on the relationship between housing price and population for CA Housing. The first column, a line graph, represents two modes recognized by MNAM between housing prices and populations. Second and third columns, heatmaps, represents changes in the activation of two modes from changes in features of interest. The first row represents changes in median income and the second row represents changes in proximity to the beach. Except for each row's feature of interest, all other remaining features have been fixed to mean values. The magnitude of the mode's activation is illustrated through the intensity of color in heatmaps. Darker blue represents higher activation of a mode. The blue color bar represents the magnitude of a mode's activation.

relationships changes from a change in other features due to their interaction terms. With this complexity, it becomes difficult to interpret those models.

Even though the MAE score improves from an increase in the complexity of models for most datasets (as expected), differences in performances among models fluctuate greatly by datasets. This could be a result of datasets having different complexity. For example, models have similar performances on the MIMIC datasets. This could be due to the datasets being too simple to not even require nonlinearity or interaction terms of models for representations. In contrast, for the ACS Income dataset, the performance increases with an increase in complexity. This could be due to the dataset being more complex and requiring nonlinearity and more interaction terms with higher degrees for models to represent the dataset well.

| | DIRECT INPUT AND OUTPUT RELATIONSHIPS | | | COMPLEX INPUT AND OUTPUT RELATIONSHIPS | |
|---|---|---|---|---|---|
| DATASETS | LR | NAM | MNAM | EBM | GBT |
| CA HOUSING | 0.54 | 0.48 | 0.46 | 0.34 | 0.31 |
| FICO | 3.38 | 2.7 | 2.7 | 2.5 | 2.4 |
| MIMIC | 1.5 | 1.5 | 1.5 | 1.5 | 1.5 |
| BIKE | 3.65 | 3.4 | 3.4 | 3.4 | 3.4 |
| ACS INCOME | 40.0 | 37.2 | 35.7 | 33.3 | 31.8 |
| ACS TRAVEL | 16.8 | 15.6 | 15.5 | 14.2 | 13.8 |
| COMPLEXITY INTERPRETABILITY | | | | | |

*Table 4.* MAE score for LR, NAM, MNAM, EBM, and GBT on CA Housing, FICO, MIMIC, BIKE, ACS Income, and ACS Travel datasets. The complexity of models increases from left to right and the interpretability of models increases from right to left.

## 4. Related Works

For interpretable models, GAM (Hastie, 2017) has been widely used. GAM transforms each feature by a function and linearly combines the transformed features, which enables features to have a fixed relationship with the output. NAM (Agarwal et al., 2021) uses neural networks while GAM uses boosted decision trees (Lou et al., 2012; Guisan et al., 2002) to transform the features. Compared to those models, MNAM has multiple outputs with probability, instead of one single estimate. These multiple outputs enable the model to represent multiple subpopulations in the dataset. Furthermore, it is more flexible for interpretation as it is able to show multiple relationships between features and labels, and how different relationships are activated by changes in a subpopulation.

To address the limitation of GAM in representing multiple subpopulations in a dataset, Generalized Additive Model with Pairwise Interactions (GA2M) (Karatekin et al., 2019) or EBM has been proposed, which adds interaction terms

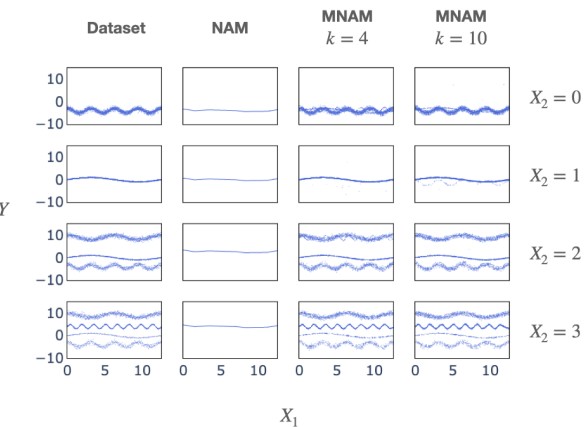

*Figure 5.* NAM versus MNAM on a dataset that has variables that identify subpopulations as observed and latent variables. The left column is a scatter plot for a dataset with different values of $X_2$. The remaining columns represent predictions from training on the dataset for NAM, MNAM with $k = 4$, and MNAM with $k = 10$. NAM clearly fails to represent the dataset as it collapses multiple relationships into one relationship. On contrary, MNAM with $k = 4$ and $k = 10$ accurately represents the dataset as it learns four relationships and activates the right ones for different values of $X_2$.

into GAM. Yet, the limitation of GA2M is that relationships between features and labels are not fixed due to its interaction terms, making the model less interpretable. The model requires users to read two graphs for interpretation. One is for a line graph on the relationship between label and feature of interest and another one is for a heatmap on interaction terms. Users have to mentally visualize the relationship by looking at two graphs to understand how the relationship changes from a change in other features. Compared to GA2M, users have to only look at one graph and don't have to mentally visualize as the relationship is fixed for MNAM.

Mixture Density Networks (MDNs) (Bishop, 1994) is the first model to use a mixture of $k$ Gaussian distributions as an outcome for a neural network. Its purpose was to solve inverse and robotics problems. MDNs is not a form of a Generalized Additive Model but more of a DNN with a mixture of $k$ Gaussian distributions as an outcome. For DNN and MDN, the relationship between a feature of interest and a label will completely change from changes in other features. It is difficult to compare all possible relationships and describe how they differ from each other.

## 5. Limitations

MNAM's current formulation is only applicable to regression problems. Unlike continuous variables, binary variables are meaningless to cluster as the only possible values are zero and one. For our future work, we will utilize different algorithms such as local interpretable model-agnostic explanations (Lime) (Ribeiro et al., 2016) to overcome such a limitation. For example, we could utilize MNAM to approximate predictions of a neural network that has been trained for the classification, as a prediction for the classification will be continuous. Using MNAM to approximate the prediction of the classification model, we will able to show multiple relationships between features and outputs and how those relationships are activated from changes in subpopulations or features.

MNAM trade-offs between the accuracy and interpretability of a model. Increasing the number of $k$ Gaussian distributions for MNAM will increase accuracy. Yet, if the number of $k$ Gaussian distributions is large, then it will be hard to interpret as there are too many possible relationships between features and outputs. The larger the number of $k$ Gaussian distributions in MNAM, the more the model will become similar to neural networks as it covers all separate relationships for all possible combinations of features. For our future works, we would explore different penalties for the number of $k$ Gaussian distributions in training to find an optimal balance between accuracy and interpretability.

Although MNAM maintains interpretability through fixed relationships between input and output, a neural network for identifying subpopulations lacks interpretability. Consequently, users must explore various inputs to determine when a certain relationship will be activated. In the future, we plan to explore alternative methods to enhance the interpretability of the neural network that is used for identifying subpopulations. This will enable users to understand when a specific relationship will be activated without the need to run the model with different inputs.

## 6. Conclusion

In this work, we introduced Mixture Neural Additive Model (MNAM), an interpretable model with more flexibility compared to GAM and NAM. While GAM and NAM have only one estimate for an output and one relationship between features and outputs, MNAM has $k$ multiple estimates for an output, with probability, and $k$ relationships between features and outputs, to represent different relationships for each potential subpopulation separately. With such advantages in flexibility, we have shown that MNAM outperforms NAM in various datasets. Furthermore, we have shown how MNAM improves interpretation by illustrating how different relationships are activated by changes in subpopulations.

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

## A. MNAM Computation Algorithm

---

**Algorithm 2** Mixture Neural Additive Models

---

**Input:** Data: $(X_1, ... X_n)$, Number of Features: $n$, Number of Gaussian Distributions: $k$, Neural Networks for Feature Transformation: $(f_1, ..., f_n)$, Neural Network for Probability: $g$

**Output:** Mixture of Gaussian Distributions: $\mathcal{N}_1(\mu_1, \sigma_1^2), ..., \mathcal{N}_k(\mu_k, \sigma_k^2), \pi_1, ..., \pi_k$

**for** $i = 1$ **to** $n$ **do**

   $\mathcal{N}_{i,1}(\mu_{i,1}, \sigma_{i,1}^2), ..., \mathcal{N}_{i,k}(\mu_{i,k}, \sigma_{i,k}^2), Z_i^\pi = f_i(X_i)$

**end for**

**for** $i = 1$ **to** $k$ **do**

   $\mu_i = \sum_{j=1}^n \mu_{j,i}$

   $\sigma_i^2 = \sum_{j=1}^n \sigma_{j,i}^2$

**end for**

$\pi_1, ..., \pi_k = g(Z_1^\pi, ..., Z_n^\pi)$

---

## B. Other Training Algorithms

### B.1. Soft-Thresholding Algorithm

Similar to the EM algorithm (Dempster et al., 1977), the ST algorithm has expectation and maximization steps for training. In the expectation step, we compute the posterior probability of subpopulations $P(Z = k|X, Y)$. As shown in Equation 2, we compute the posterior probability by utilizing Bayesian Theorem,

$$
\begin{aligned}
P(Z = k|X, Y) &= \frac{P(X, Y|Z = k)P(Z = k)}{P(X, Y)} \\
&= \frac{P(X, Y|Z = k)P(Z = k)}{\sum_{i=1}^k P(X, Y|Z = k)P(Z = k)},
\end{aligned}
\tag{2}
$$

where $P(X, Y|Z = k)$ is the likelihood of $k$th Gaussian distribution for the giveninput, and $P(Z = k)$ is the prior probability of a subpopulation, which is predicted from MNAM. In the maximization step, we update the weights of MNAM to maximize the expectation or posterior probability of the subpopulations. First, we compute GNLL losses for all Gaussian distributions, and then GNLL losses for all the Gaussian distributions are linearly combined with weights matching posterior probabilities from the expectation step. This ensures weights used to compute Gaussian distribution with a higher likelihood are updated more. Cross-entropy loss between the prior probability predicted from MNAM and the posterior probability computed in the expectation step is computed with a similar purpose as in the HT algorithm. Algorithm 3 summarizes the proposed training algorithm.

---

**Algorithm 3** Soft-Thresholding (ST) Algorithm

---

**Input: Data** $(X, Y)$**, MNAM** $f$**, GNLL loss** $g$**, Crossentropy loss function** $h$**, Rate for cross-entropy loss** $\lambda$

$\mathcal{N}_1(\mu_1, \sigma_1), ..., \mathcal{N}_k(\mu_k, \sigma_k), \pi_1, ..., \pi_k = f(X)$

**for** $i = 1$ **to** $k$ **do**

   $gau\_loss_i = g(\mathcal{N}_i(\mu_i, \sigma_i), Y)$

   $gau\_like_i = p(Y; \mu_i, \sigma_i)$

**end for**

$mar\_prob = \sum_{j=1}^k gau\_like_j \cdot \pi_j$

$\hat{\pi}_1, ..., \hat{\pi}_k = \dfrac{gau\_like_1 \cdot \pi_1}{mar\_prob}, ..., \dfrac{gau\_like_k \cdot \pi_k}{mar\_prob}$

$gau\_loss = \sum_{i=1}^k gau\_loss_i \cdot \hat{\pi}_i$

$prob\_loss = h((\pi_1, ..., \pi_k), (\hat{\pi}_1, ..., \hat{\pi}_k))$

$total\_loss = gau\_loss + \lambda \cdot prob\_loss$

---

## B.2. Comparison of Training Algorithms

For comparing HT and ST algorithms, we evaluated numerical stability (NS), computation of time (CT), and accuracy. Using the dataset from the pedagogic study, we trained MNAM with different learning rates 20 times each to evaluate metrics. NS was assessed by computing the percentage of successful training without exploding gradient. CT was assessed by tracking average training time in seconds. Accuracy was assessed by computing MAE and EMD on the test set. Table 5 shows the evaluation of those metrics.

The HT algorithm had better performance in NS and CT. One of the explanations for better performance in NS is that the HT algorithm only passes minimum GNLL loss while the ST algorithm passes all GNLL losses with weights for an update. The ST algorithm passes more loss compared to the HT algorithm, which makes it numerically unstable during training. Furthermore, the ST algorithm has higher CT compared to the HT algorithm because it requires more computation to estimate the posterior probability, the HT algorithm only needs to find a minimum GNLL loss for training. For accuracy, the HT algorithm had a higher EMD score and lower MAE score compared to the ST algorithm. Based on the priority of two metrics, one could choose one algorithm over the other. For this study, we used the HT algorithm due to its better performance in NS and CT.

| | HARD-THRESHOLDING ALGORITHM | | | | SOFT-THRESHOLDING ALGORITHM | | | |
| LR | NS | CT | MAE | EMD | NS | CT | MAE | EMD |
|---|---|---|---|---|---|---|---|---|
| 0.05 | 100% | 217.61 | 43.89 | 145.38 | 0% | NA | NA | NA |
| 0.01 | 100% | 386.13 | 5.23 | 4.03 | 0% | NA | NA | NA |
| 0.005 | 100% | 470.94 | 3.12 | 0.25 | 0% | NA | NA | NA |
| 0.001 | 100% | 462.05 | 3.14 | 0.19 | 95% | 488.77 | 2.82 | 0.40 |
| 0.0005 | 100% | 929.62 | 3.09 | 0.19 | 100% | 891.53 | 2.98 | 0.36 |
| 0.0001 | 100% | 1029.1 | 3.12 | 0.29 | 100% | 1036.5 | 3.06 | 0.30 |
| $5e^{-05}$ | 100% | 992.03 | 3.31 | 0.28 | 100% | 1050.7 | 3.08 | 0.45 |

*Table 5.* Comparision of HT algorithm and ST algorithm on data from pedagogic study

# C. Table of optimized parameters for MNAM

| DATASET | LEARNING RATE | WEIGHT DECAY | OUTPUT PENALTY | NUMBER OF GAUSSIAN distribution | CROSS-ENTROPY LOSS |
|---|---|---|---|---|---|
| CA HOUSING | 0.009896 | 3.8512E-05 | 0.03363 | 2 | 0.6118 |
| FICO | 0.02757 | 6.6649E-05 | 0.006145 | 6 | 0.4718 |
| MIMIC | 0.01805 | 7.0946E-05 | 0.01908 | 2 | 0.7214 |
| BIKE | 0.01172 | 9.1022E-05 | 0.09256 | 6 | 0.3537 |
| ACS INCOME | 0.02873 | 9.13E-05 | 0.00167 | 4 | 0.494 |
| ACS TRAVEL | 0.01894 | 9.3377E-05 | 0.0028 | 4 | 0.3634 |

*Table 6.* Optimized parameters for MNAM on six datasets

# D. Training Effeciency

Table 7 shows the comparison of average training time, training time per epoch, and the number of epochs required to train NAM and MNAM on different datasets. As expected, MNAM takes longer to train per epoch than NAM because it has an additional neural network for computing mode weights. However, interestingly, MNAM was faster than NAM in training for half of the datasets, and on average, MNAM required fewer epochs than NAM for all datasets except one. Our hypothesis is that MNAM's assignment of modes to subpopulations effectively shrinks the space for the model to explore, resulting in fewer epochs needed for training, as each mode only needs to represent one subpopulation. In contrast, NAM has only one outcome that must represent multiple subpopulations, causing it to oscillate among subpopulations for representation during training. Overall, MNAM may be more efficient than NAM in terms of training time, despite having more parameters to compute.

| DATASET | TRAINING TIME (SECONDS)↓ | | TRAINING TIME PER EPOCH (SECONDS)↓ | | NUMBER OF EPOCHS↓ | |
|---|---|---|---|---|---|---|
| | NAM | MNAM | NAM | MNAM | NAM | MNAM |
| CA HOUSING | 636.38 | 411.73 | 0.72 | 0.84 | 885.36 | 489.85 |
| FICO | 326.26 | 337.54 | 0.78 | 0.85 | 417.34 | 397.67 |
| MIMIC | 173.10 | 278.58 | 0.93 | 1.09 | 186.64 | 256.53 |
| BIKE | 410.42 | 477.62 | 1.37 | 1.85 | 298.76 | 258.36 |
| ACS INCOME | 1460.31 | 896.41 | 3.22 | 4.47 | 453.65 | 200.40 |
| ACS TRAVEL | 2411.51 | 946.55 | 3.86 | 5.75 | 624.43 | 164.58 |

*Table 7.* Comparisons of average training time in seconds, training time per epoch in seconds, and number of epochs between MNAM and NAM

## E. Table of Earth Mover's Distance score for models

| DATASETS | DIRECT INPUT AND OUTPUT RELATIONSHIPS | | | COMPLEX INPUT AND OUTPUT RELATIONSHIPS | |
|---|---|---|---|---|---|
| | LR | NAM | MNAM | EBM | GBT |
| CA HOUSING | 0.29 | 0.24 | 0.077 | 0.11 | 0.09 |
| FICO | 1.16 | 0.73 | 0.60 | 0.51 | 0.51 |
| MIMIC | 1.36 | 1.43 | 0.24 | 1.32 | 1.25 |
| BIKE | 3.06 | 2.50 | 0.26 | 2.45 | 2.43 |
| ACS INCOME | 27.1 | 21.3 | 7.4 | 14.5 | 12.9 |
| ACS TRAVEL | 14.2 | 12.8 | 3.1 | 8.9 | 8.7 |

COMPLEXITY
INTERPRETABILITY

*Table 8.* EMD score for LR, NAM, MNAM, EBM, and GBT on CA Housing, FICO, MIMIC, BIKE, ACS Income, and ACS Travel datasets. The complexity of models increases from left to right and the interpretability of models increases from right to left.

# F. Relationships plots on other datasets

## F.1. MIMIC

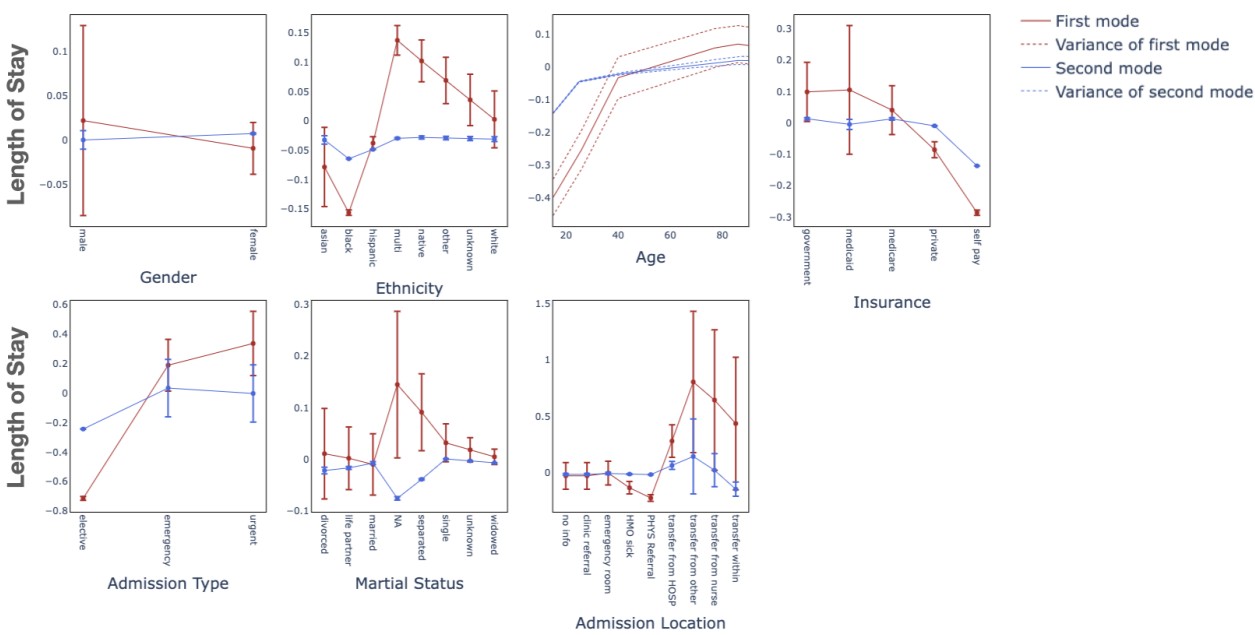

*Figure 6.* Learned relationships between features and labels for the MNAM on MIMIC datasets. Solid lines represent the mean of the relationships and dotted lines represent their uncertainties.

## F.2. Housing

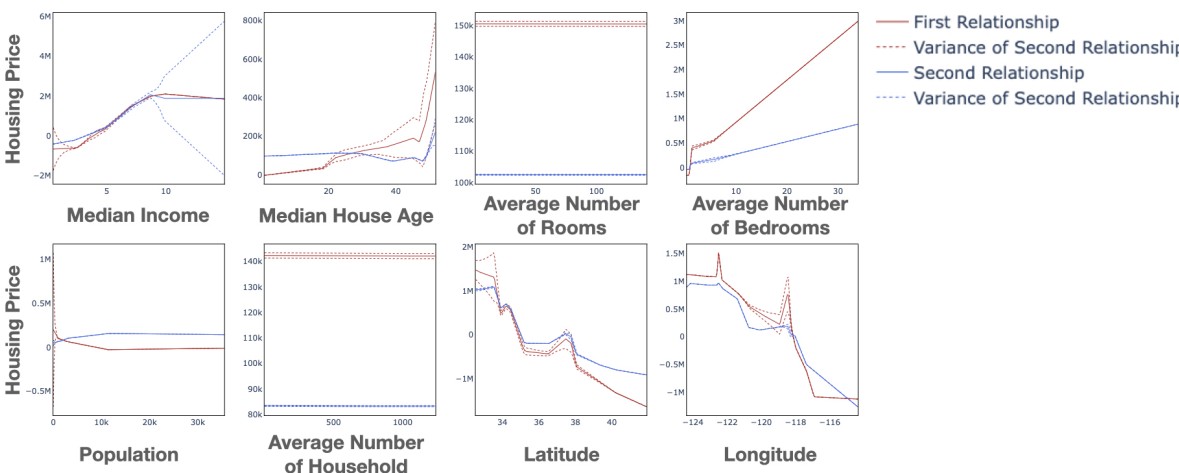

*Figure 7.* Learned relationships between features and labels for the MNAM on Housing datasets. Solid lines represent the mean of the relationships and dotted lines represent their uncertainties.

## F.3. FICO

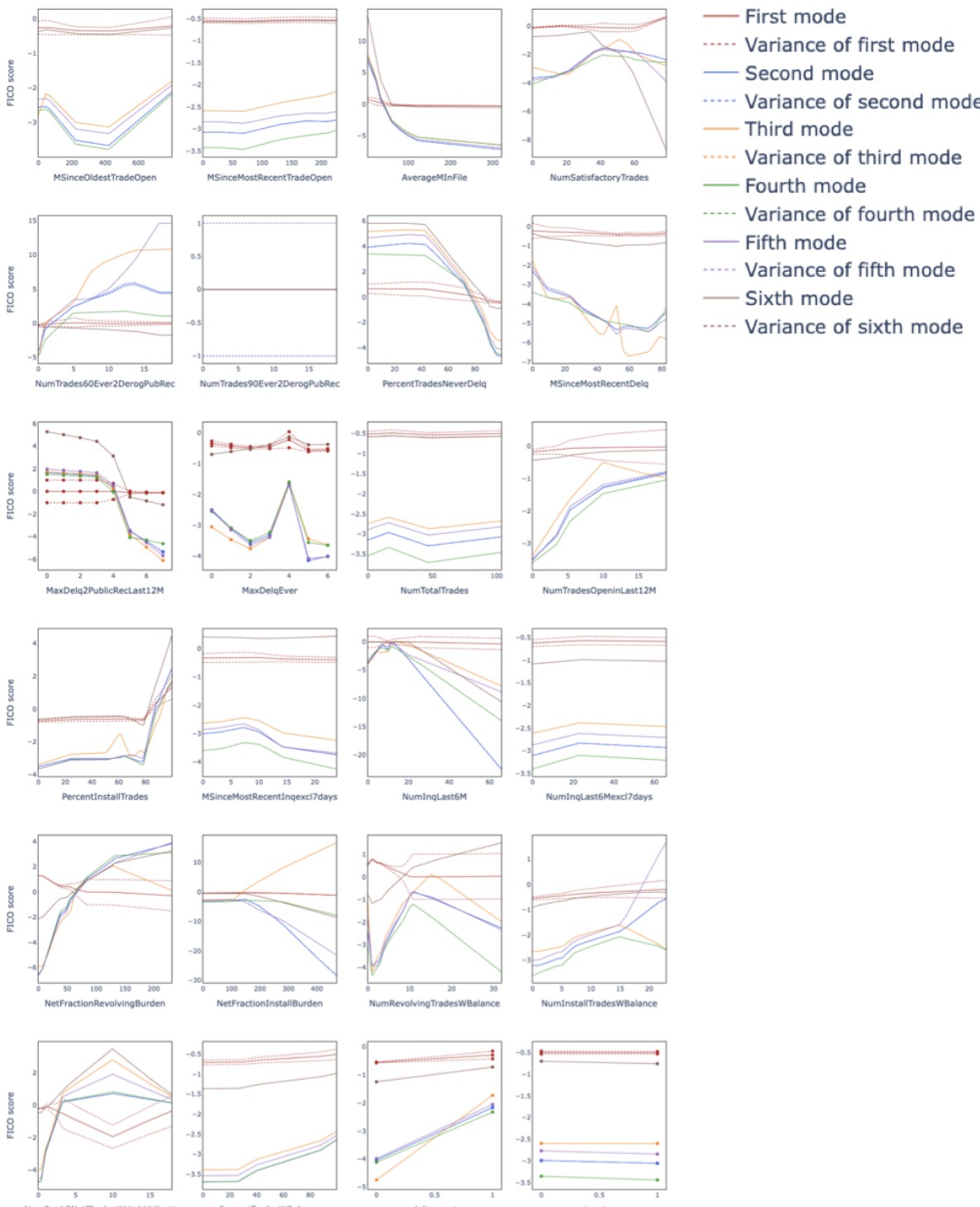

*Figure 8.* Learned relationships between features and labels for the MNAM on FICO datasets. Solid lines represent the mean of the relationships and dotted lines represent their uncertainties.

## F.4. BIKE

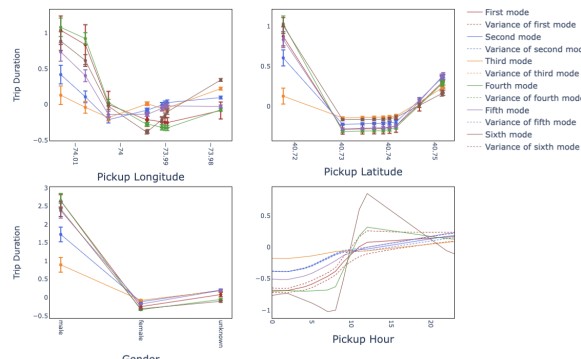

*Figure 9.* Learned relationships between features and labels for the MNAM on BIKE datasets. Solid lines represent the mean of the relationships and dotted lines represent their uncertainties.

## F.5. ACS Income

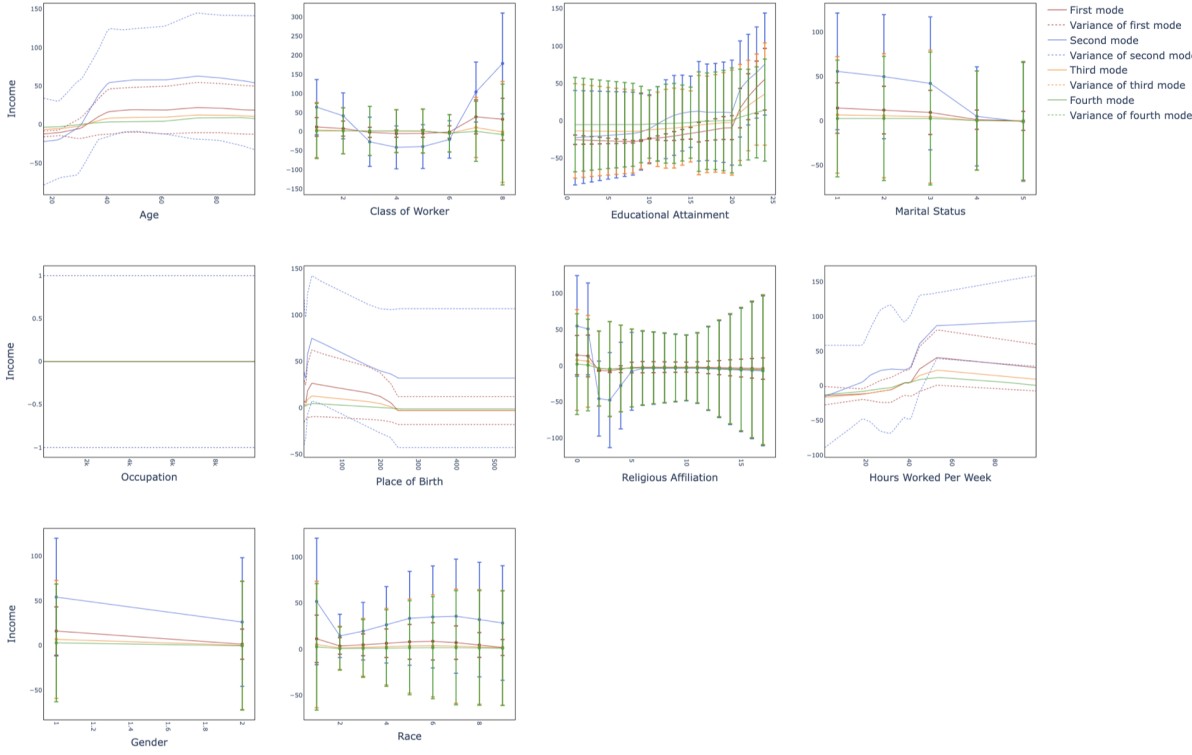

*Figure 10.* Learned relationships between features and labels for the MNAM on ACS Income datasets. Solid lines represent the mean of the relationships and dotted lines represent their uncertainties.

## F.6. ACS Travel

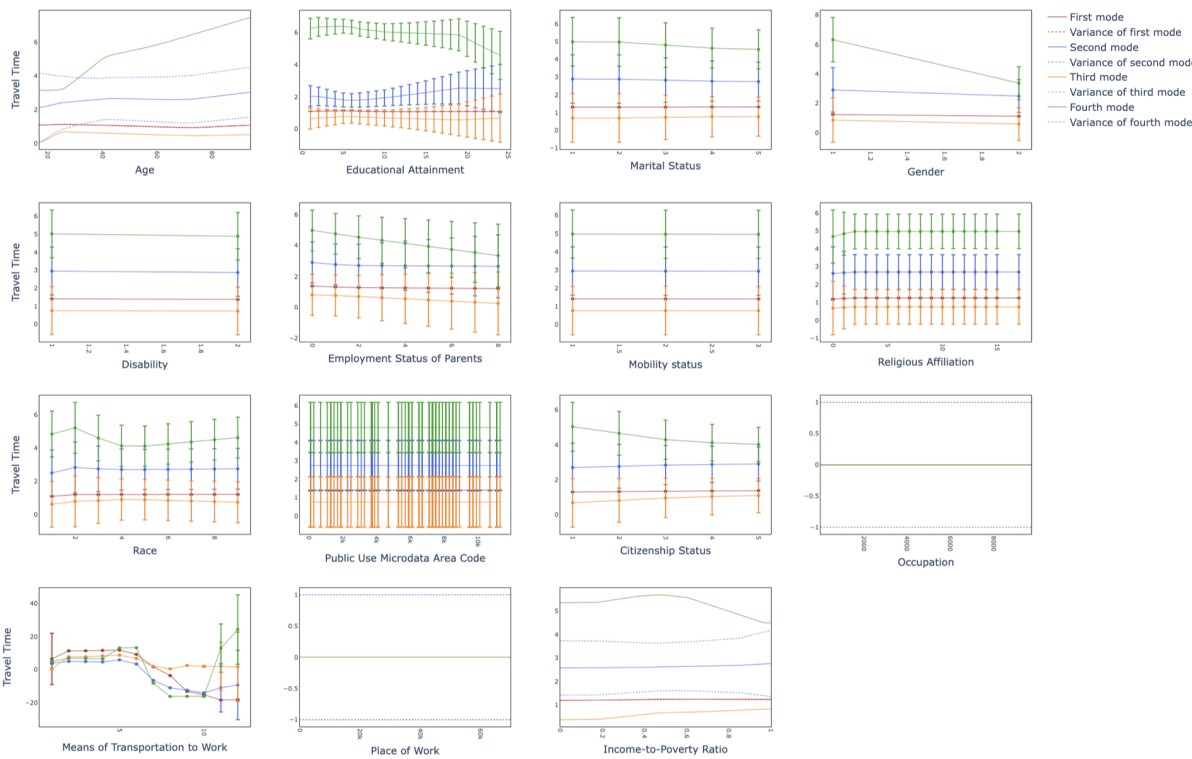

*Figure 11.* Learned relationships between features and labels for the MNAM on ACS Travel datasets. Solid lines represent the mean of the relationships and dotted lines represent their uncertainties.

