# OpenReview forum: "Generalizing Neural Additive Models via Statistical Multimodal Analysis"
_ICML.cc/2023/Workshop/IMLH — IMLH 2023 Poster_

### Official Review · Reviewer_Sh35 · 2023-06-09
**The paper proposes a Mixture of Neural Additive Models (MNAM) to address the limitations of Generalized Additive Models (GAM) and Neural Additive Models (NAM) when handling data with multiple distinct subpopulations. MNAM maintains the interpretability of GAM and NAM, but improves their adaptability by learning relationships in a multimodal fashion, and activating the appropriate relationships based on identified subpopulations.**

**Rating:** 6
**Confidence:** 3

**Review:**

Pros:

Addressing an Overlooked Limitation: The paper addresses a significant, yet overlooked, limitation in GAM and NAM, providing an important contribution to the field of machine learning interpretability.

Proposing a Novel Solution: The Mixture of Neural Additive Models (MNAM) is an innovative solution that balances interpretability and adaptability, particularly for complex datasets with multiple distinct subpopulations.

Multimodal Relationships: MNAM learns and represents multiple relationships between features and outputs, enhancing its ability to handle complex data structures. The model outperforms NAM in various datasets, showcasing its effectiveness and robustness.

Maintains Interpretability: Despite its complexity, MNAM provides interpretability  how different relationships are activated by changes in subpopulations.

Extensive Empirical Evaluation: The paper includes numerous empirical observations and studies to demonstrate the effectiveness and benefits of the MNAM.

Robust Results: The results generated by MNAM across different datasets attest to the quality of the proposed method.

Cons:

Limited Scope: MNAM's current formulation is only applicable to regression problems, restricting its use in other types of tasks like classification.

Interpretability Trade-off: With an increase in the number of Gaussian distributions (k), interpretability may become a challenge due to the presence of too many possible relationships between features and outputs.

Accuracy-Interpretability Balance: The model currently faces a trade-off between improving accuracy (by increasing the number of Gaussian distributions) and maintaining interpretability.

Need for Future Developments: MNAM still requires further improvements and extensions to overcome its limitations, such as its applicability to classification tasks and finding the optimal balance between interpretability and accuracy.

Complex Concepts: While the authors have made efforts to explain their model, some concepts, particularly around the choice and implications of the number of Gaussian distributions, might require more clarity for a broader audience.

Extension of Existing Models: While MNAM is a new approach, it is essentially an extension of existing models like GAM and NAM. It introduces multimodal learning but within the broad framework of additive models, which might not be seen as highly original by some.

---

### Official Review · Reviewer_9VMx · 2023-06-13
**An interesting extension to neural additive models**

**Rating:** 7
**Confidence:** 3

**Review:**

This paper propose MNAM, a extension to NAM that considers multiple possible relationships between features and outputs, to address the problem that NAM model may incorrectly collapse those relationships by fitting in a unimodal fashion. Experiments demonstrate that MNAM achieves better performance than NAM and MNAM is able to capture the multiple relationships between features and outputs.

The motivation of the paper is well-grounded, and the proposed extension is interesting and straightforward.

However, the interpretability of MNAM can be further analyzed. For example, how do we interpret the multiple relationships - when will the first relationship hold and when will the second relationship hold? Also, the organization of the paper can be improved. For examples, the method section would be better to start with problem definition instead of detailed method description, and the training details would be better to included in the experiment section, instead of mixed in the method section.

---

### Official Review · Reviewer_eYtQ · 2023-06-18
**Extension of NAMs to muli-modal settings through mixture density networks. Impressive gains. Long paper**

**Rating:** 7
**Confidence:** 5

**Review:**

This paper extends the NAM framework to multi-modal settings by using mixture density networks (MDNs).
The core idea is to utilize MDNs which are neural networks that produce outcomes represented as a combination of k Gaussian distributions. These MDNs then serve as linking functions for generalized additive models (GAMs) to capture the connection between input variables and outcomes. I really like the toy example in figure 1 which clearly highlights the current limitations of Linear models and GAMs in modeling complx non-linear relationships. This extension is straight-forward but an important contribution. Furthermore there is an impressive gains as measured through the likelihood score while there is not a significant improvement in task-specific performance measures such as MAE. However, this method leads to more interpretable results. I would like to see a discussion on the increased computational complexity of MNAM. Furthermore, intuitively MNAMs should provide more robust results when tested on samples drawn from a different distribution (i.e, domain shift). I highly encourage the authors to consider such examples to demonstrate the benefits of MNAMs over NAMs, such as the datasets provided here to test for domain generalization (https://arxiv.org/abs/2210.05775) in regression tasks.

---

### Meta-Review · Area_Chair_3DGx · 2023-06-18

**Recommendation:** Accept (Poster)
**Confidence:** 4

**Metareview:**

This paper extends the NAM framework to multi-modal settings by using mixture density networks (MDNs). The paper has been positively reviewed, with the reviewers finding it well-motivated and well-written. In addition, the proposed methods have been deemed novel and timely. All reviewers recommended acceptance.

There are a few concerns raised by the reviewers about interpretability and further analysis, which the authors should take into account when preparing for the camera ready version.

---

### Decision · Program_Chairs · 2023-06-20

Accept (Poster)